# Impact of Whole-Genome Sequencing of *Mycobacterium tuberculosis* on Treatment Outcomes for MDR-TB/XDR-TB: A Systematic Review

**DOI:** 10.3390/pharmaceutics15122782

**Published:** 2023-12-15

**Authors:** Druti Hazra, Connie Lam, Kiran Chawla, Vitali Sintchenko, Vijay Shree Dhyani, Bhumika T. Venkatesh

**Affiliations:** 1Department of Microbiology, Kasturba Medical College, Manipal, Manipal Academy of Higher Education, Manipal 576104, Karnataka, India; druti.hazra@learner.manipal.edu; 2Sydney Institute for Infectious Diseases, Faculty of Medicine and Health, University of Sydney, Sydney, NSW 2006, Australia; connie.lam@health.nsw.gov.au; 3Centre for Infectious Diseases and Microbiology-Public Health, Westmead Hospital, Westmead, Sydney, NSW 2145, Australia; 4Kasturba Medical College, Manipal, Manipal Academy of Higher Education, Manipal 576104, Karnataka, India; vs.dhyani@manipal.edu; 5Public Health Evidence South Asia, Prasanna School of Public Health, Manipal Academy of Higher Education, Manipal 576104, Karnataka, India; bhumikatv@campbellcollaboration.org

**Keywords:** tuberculosis, drug resistance, whole-genome sequencing, drug therapy, drug treatment, mycobacterium tuberculosis

## Abstract

The emergence and persistence of drug-resistant tuberculosis is a major threat to global public health. Our objective was to assess the applicability of whole-genome sequencing (WGS) to detect genomic markers of drug resistance and explore their association with treatment outcomes for multidrug-resistant/extensively drug-resistant tuberculosis (MDR/XDR-TB). Methods: Five electronic databases were searched for studies published in English from the year 2000 onward. Two reviewers independently conducted the article screening, relevant data extraction, and quality assessment. The data of the included studies were synthesized with a narrative method and are presented in a tabular format. Results: The database search identified 949 published articles and 8 studies were included. An unfavorable treatment outcome was reported for 26.6% (488/1834) of TB cases, which ranged from 9.7 to 51.3%. Death was reported in 10.5% (194/1834) of total cases. High-level fluoroquinolone resistance (due to *gyrA* 94AAC and 94GGC mutations) was correlated as the cause of unfavorable treatment outcomes and reported in three studies. Other drug resistance mutations, like kanamycin high-level resistance mutations (*rrs* 1401G), *rpoB* Ile491Phe, and *ethA* mutations, conferring prothionamide resistance were also reported. The secondary findings from this systematic review involved laboratory aspects of WGS, including correlations with phenotypic DST, cost, and turnaround time, or the impact of WGS results on public health actions, such as determining transmission events within outbreaks. Conclusions: WGS has a significant capacity to provide accurate and comprehensive drug resistance data for MDR/XDR-TB, which can inform personalized drug therapy to optimize treatment outcomes.

## 1. Introduction

Despite global efforts to eliminate tuberculosis disease, drug-resistant tuberculosis (DR-TB) remains a major challenge for tuberculosis (TB) eradication programs. The World Health Organization (WHO) reported 410,000 new cases of multi-drug-resistant TB (MDR-TB)/rifampicin-resistant TB (RR-TB) in 2022 [1]. Among these cases, 17% were *Mycobacterium tuberculosis* (MTB) isolates resistant to two first-line drugs, namely, isoniazid and rifampicin, are defined as MDR-TB; MDR-TB together with resistance to a fluoroquinolone and another group A antitubercular medication (e.g., linezolid or bedaquiline) are considered extensively drug-resistant TB (XDR-TB) and are considered difficult to treat due to limited effective drug therapies [2,3,4]. The detection of DR-TB using culture-based drug susceptibility testing (DST) can be challenging. It remains time consuming and expensive and requires a high-capacity laboratory, as well as methodological standardization of testing for second-line antimycobacterial agents [5,6]. In 2019, only 71% of RR/MDR-TB cases were tested for second-line drug fluoroquinolones, which could have been due to the inaccessibility of test facilities or a culture of slow-growing MTB for phenotypic DST [1].

At present, there are PCR-based genotypic assays, which are able to rapidly identify common drug resistance markers making them advantageous over time-consuming phenotypic DST. However, these diagnostic methods can only detect a limited number of genetic mutations already known to confer drug resistance [7,8]. In contrast, whole-genome sequencing (WGS) has the ability to interrogate the entire genome sequence of a bacterium and enables the detection of multiple novel mutations that may be associated with drug resistance. Depending on the sequencing platform and culture method used to grow MTB, WGS resistomes can be achieved in as little as 7 days and have the capacity to identify a broad range of genetic polymorphisms, which can help to develop new treatment strategies and improve the case management for DR-TB [8,9]. Thus, many of the limitations of PCR-based diagnostic and drug resistance detection methods can be overcome by WGS to provide genome-wide insight into emerging clinically relevant mutations in the MTB [10,11].

The expansion of WGS-based applications into TB diagnostics was documented in recent studies that demonstrated the applicability of WGS in the clinical management of tuberculosis and explored the different perspectives on WGS utility and accuracy [10,12,13]. For instance, Hatherell et al. [14] emphasized that epidemiological and clinical data coupled with WGS results can strengthen investigations of TB transmission; this finding is further supported by Van Der Werf et al. [15], who illustrated the advantages of WGS in identifying clusters, tracking transmission direction, and investigating international TB outbreaks. From a drug resistance perspective, Papaventsis et al. [8] concluded that WGS could be a promising alternative to existing DST methods, while Nieto Ramirez et al. [16] highlighted the crucial role of WGS in discovering and confirming new drug resistance mutations. Understanding and detecting baseline drug resistance has also been recognized as a major risk factor that impacts treatment efficacy in patients with a prior history of TB treatment and can therefore develop MDR/XDR-TB [17].

Improvements in WGS technology and analysis have enabled earlier and more accurate detection of drug resistance in TB cases, as well as guide efforts to reduce TB transmission. However, as highlighted by Papaventsis et al. [8], there is currently a lack of knowledge on how TB-WGS results correlate with clinical outcomes. This review was therefore conducted to address the current evidence gap that exists on whether the results of TB-WGS affect treatment outcomes in TB patients. We evaluated the current literature to determine whether the implementation of TB-WGS improves clinical outcomes and whether WGS is able to support TB control programs. We further focused on the ability of WGS to detect genomic markers of drug resistance and determine their association with treatment outcomes for MDR/XDR-TB. The implementation of WGS in routine TB diagnostics could significantly improve clinical outcomes and aid TB control programs, and it is hoped that this review will provide insight into the design of future WGS studies that incorporate treatment outcomes with WGS results. 

## 2. Materials and Methods

The Preferred Reporting Items for Systematic Reviews and Meta-Analyses (PRISMA) guidelines were followed to develop this systematic review (Appendix A). This protocol of this review was registered with the International Prospective Register of Systematic Reviews (PROSPERO) database (registration ID: CRD42020197099).

Eligibility criteria: We included any observational or experimental studies that described different applications of WGS to genotypic resistance testing for TB and detection of MDR-TB/XDR-TB, as well as treatment outcomes of the patients. The results were limited to original English-language research articles published from the year 2000 onward. Letters to the editor, case studies, short abstracts, conference papers, review articles, and comments were not included in this study. There were no geographical restrictions applied, as the review was designed to include global data.

Search strategy: Five electronic databases were used to conduct a literature search, i.e., Scopus, Cochrane Library, Web of Science, NCBI PubMed (MEDLINE), and CINAHL (Ebsco). A structured search strategy was employed based on the following keywords: “drug-resistant tuberculosis”, “tuberculosis”, “treatment outcome”, and “whole genome sequencing”. Also, site-specific biomedical terms, like Medical Subject Headings (MeSH), and Emtree were included, and Boolean operators were used to combine all search terms (Appendix A).

Selection of studies: The search results retrieved from the electronic databases were imported into Zotero and duplicate studies were removed. For the title–abstract screening phase, the retrieved studies were exported to a Microsoft Excel spreadsheet (Microsoft, 2019). Two reviewers (D.H. and C.L.) independently assessed the abstracts using the eligibility criteria described previously. Studies identified as relevant by at least one of the reviewers were included in the full-text assessment. The full text for each study was retrieved and assessed by two reviewers (D.H. and C.L.) independently. Any disagreement between the reviewers’ interpretations was discussed between them, and if not resolved, was clarified by a third reviewer. The final decision of including/excluding a study was confirmed by all reviewers and appropriately documented.

Data extraction: Pertinent data were extracted separately by the primary reviewers from each included article. A standardized data extraction spreadsheet was developed after a pilot assessment. The key data extracted included the following: basic descriptors (i.e., names of authors, year of publication, journal, the country where the patients were enrolled in the study, duration of the study, study design, study setting, method of WGS employed in the study), study participants (number of patients included, age, gender, risk groups when available), main findings (phenotypic DST and WGS, number of MDR-TBs/XDR-TBs identified, drug-resistance-associated mutations detected using WGS, technical details about diagnostic tools), and treatment outcomes (treatment completion, disease cure, death, treatment default, TB relapse).

Assessment of quality: Two reviewers (D.H. and C.L.) assessed the quality of the included studies independently using the Quality Assessment Tool for Observational Cohort and Cross-Sectional Studies of the National Heart, Lung, and Blood Institute of the National Institutes of Health (NHLBI) [18]. The quality assessment tool was developed based on fourteen parameters to assess the internal validity of the observational cohort and cross-sectional studies. This tool comprises questions to assess the possible shortcomings in the methodology of a study, such as the risk of potential bias, confounding, power of the study, the strength of the causal relationship between the exposure and outcome, and other aspects. During the quality evaluation process, reviewers select the options of yes/no/not applicable (NA)/not reported (NR)/cannot determine (CD) responses for each question. Finally, the reviewers assessed the quality of each study based on a rating system of good, fair, or poor [18].

Data analysis: We performed a narrative data synthesis of the included studies and presented them in a tabular form. The meta-analysis was not performed due to the high levels of data heterogeneity among the included studies.

## 3. Results

### 3.1. Selected Sources

The database search identified 949 published articles. After excluding 329 duplicates, 620 title–abstracts were screened and 49 articles were selected for the full-text examination. Finally, 8 eligible studies out of 49 articles were included in the narrative synthesis. The study selection process and outcomes are illustrated in Figure 1.

Of these eight manuscripts, four (50%) were prospective/observational cohorts [11,19,20,21] and two (25%) were prospective observational studies [17,22]. Out of the two remaining studies, one (12.5%) was based on a retrospective cohort population [23] and the other was a retrospective observational study [24]. In this review, the single-centric studies included samples from five countries: China (*n* = 2) [17,24], South Africa (*n* = 1) [22], Tanzania (*n* = 1) [11], Bangladesh (*n* = 1) [23], and Uganda (*n* = 1) [20]. Two of the studies [19,21] included multiple sites. Full details of the included studies are summarized in Table 1, Table 2 and Table 3.

The sample sizes of the included studies ranged from 41 to 582, including a total of 2076 tuberculosis-positive patients. Of the total number of patients, 67.5% (1097/1623) were males. A total of 1313 MDR-TB cases (range: 24–449 cases) and 27 XDR-TB cases (range: 3–24 cases) from the eight studies were recorded in this review.

### 3.2. Whole-Genome Sequencing and Analysis of Drug Resistance Mutations

Illumina (Illumina, San Diego, CA, USA) short-read sequencing was the most commonly used (seven out of eight studies) sequencing platform for whole-genome sequencing [11,17,20,21,22,23,24], and one study applied the PyroMark Q96 ID system (Qiagen, Valencia, CA, USA) [19]. Genotypic PCR-based confirmation of drug resistance mutations was performed in two studies [21,22]. Makhado et al. [22] and Zurcher et al. [21] both used Xpert MTB/RIF (Cepheid) and GenoType MTBDRplus to detect rifampicin and isoniazid resistance. To identify the Ile491Phe-bearing isolates, the authors further performed multiplex allele-specific PCR and Sanger sequencing of the *rpoB* and Deeplex-MycTB (Genoscreen) targeted deep sequencing methods [22]. 

Bioinformatic analysis and prediction of drug resistance or drug susceptibility using open-source pipelines were described in seven studies [11,17,20,21,22,23,24], compared with one study that employed IdentiFire software (Qiagen, Valencia, CA, USA) [19]. Sequencing reads that met quality control criteria for each study were aligned to the *M. tuberculosis* H37Rv reference genome using bioinformatic programs, including BCFtools [21], bowtie2 [17,22,24], BWAmem/BWA [11,20], and TBProfiler [23]. Further variant calling and identification of SNPs or other genomic mutations were performed with SAMtools/SAMtools mpileup2 [17,20,21,22,24], BCFtools [20], Velvet [20], TBProfiler [11,21,23], or LoFreq [11].

Confirmatory phenotypic DST was carried out using either Löwenstein–Jensen media (five out of eight) or MGIT (Becton Dickinson) liquid culture in six out of eight studies. None of the studies explicitly stated that WGS information was used to guide or alter TB treatment.

### 3.3. Drug Resistance and Treatment Outcomes

Tuberculosis patients were followed up for an average of 14.9 months, ranging between 6 and 27 months. Treatment outcomes for 1834 (88.3%) cases were recorded in the overall 2076 TB cases. Favorable treatment outcomes were reported for 70.9% (1301/1834) TB cases, whereas 26.6% (488/1834) cases had unfavorable treatment outcomes, which ranged from 9.7% to 51.3% (Table 1, Table 2 and Table 3). Death was reported in 10.5% (194/1834) of the TB cases and 5.5% (100/1834) had suffered treatment failure.

High-level fluoroquinolones (FQs) resistance (due to *gyrA* 94AAC and 94GGC mutations) was correlated as the cause of unfavorable treatment outcomes and reported in three studies [17,19,23]. Other drug resistance mutations, including a kanamycin (KAN) high-level resistance mutation (*rrs* 1401G) [19], Ile491Phe mutation of *rpoB* conferring resistance to rifampicin (RIF) [22], and mutations in the *ethA* conferring prothionamide (PTO) resistance [17], were also associated with poor treatment outcomes. However, not all studies provided multivariable-adjusted effect estimates of mortality risk associated with drug-resistance-associated genes. He et al. also reported the link between phenotypic resistance to Cycloserine (Cs) and high-level resistance to moxifloxacin (MXF) with adverse outcomes [17]. Lumpens et al. documented poor clinical outcomes in cases with high-level fluoroquinolone resistance (phenotypic DST) and isoniazid high-level resistant TB (both phenotypic DST and genotypic DST) [23]. 

### 3.4. Whole-Genome Sequencing and Detection of Undertreated TB Cases

WGS resistomes were used by Zurcher et al. to retrospectively evaluate whether appropriate treatment was provided. The reported accordance between conventional phenotypic DST and WGS was 80% for pan-susceptible, 8% for monoresistant, 66% for MDR, and 33% for pre-XDR or XDR tuberculosis. Based on the WGS drug resistance report, Zucher et al. found that 15% (77/530) of study participants were treated inappropriately, among which 11% (60/530) were undertreated and 3% (17/530) were overtreated [23]. 

The odds of death in undertreated TB cases were 4.92 (95% CI 2.47–9.78) compared with patients receiving appropriate treatment. Chen et al. also concluded that the administration of an inappropriate treatment regimen leads to treatment failure in MDR-TB cases and insufficient treatment regimens induce acquired drug resistance in primarily sensitive MTB strains [24].

### 3.5. Secondary Outcomes

WGS has the potential to provide additional benefits beyond detecting drug resistance, as noted in the selected studies. All eight studies performed phenotypic DST in addition to WGS; correlations between WGS resistome and phenotypic DST results were included in three studies [11,21,23]. Katale et al. reported a strong agreement between WGS and phenotypic DST for rifampicin (97%), isoniazid (81%), and streptomycin (95%), as well as with Xpert MTB/RIF for the detection of rifampicin resistance (95%) (kappa = 1.00) [11]. Makhado et al. [22] detected the transmission of MDR-TB with a *rpoB* Ile491Phe mutation using WGS, which remained undetected by the routine diagnostic tools. Furthermore, the authors measured the genomic distances based on SNP variation and showed that two lineages (4.4.1.1 and 4.1.1.3) with *rpoB* Ile491Phe had emerged independently [22]. Similarly, Clark et al. also demonstrated the utility of WGS in monitoring the emergence of drug resistance and transmission of MDR-TB in the community [20]. Chen et al. concluded that WGS is advantageous for the monitoring of reinfection by another MDR-TB, acquired resistance, and low-frequency mutated resistance genes with high sensitivity [24]. Only one study considered the cost and turnaround time of performing WGS. He et al. concluded that WGS could be more cost-effective from a laboratory perspective than phenotypic DST, with an average of USD 60 for WGS compared with USD 40 for first-line phenotypic DST (solid media) and at least USD 90 for all second-line drugs in a liquid medium; however, population-level cost-effectiveness studies are required from healthcare and societal perspectives to determine whether the availability of WGS resistomes decreases the overall costs to society of treating TB disease. Findings from this study also showed that when compared with phenotypic DST, WGS from MTB clinical isolates had a shorter turnaround time, i.e., 7 days vs. 30 days (range: 26–32 days) for solid media and 10 days (range 4–13 days) for liquid media [17].

### 3.6. Quality Assessment

Based on the NHLBI Quality Assessment Tool for Observational Cohort and Cross-Sectional Studies, only one (12.5%) study was rated as good quality, whereas five (62.5%) were rated as fair quality, and two studies (25%) were considered poor quality (Table 4).

## 4. Discussion

This study attempted to correlate the capability of the WGS resistome of MTB to predict drug-resistance-associated genes that lead to poor treatment outcomes in MDR/XDR-TB. In this systematic review, we summarized the findings of eight independent observational studies. The proportion of cases with unfavorable outcomes ranged from 9.7 to 51.3% and WGS detected several mutations conferring clinically relevant drug resistance, such as mutations in the *gyrA* for high-level fluoroquinolone resistance, *rrs* for high-level kanamycin resistance, *rpoB* Ile491Phe conferring rifampicin resistance harbored in MDR-TB, and *ethA* for prothionamide resistance, which were all associated with poor outcomes.

A cohort study from Peru performed FQ-targeted gene sequencing using molecular inversion probes and concluded that a high-level FQ resistance significantly predicted poor treatment outcomes [25]. Mutations in the quinolone-resistance-determining region (QRDR) of DNA gyrase subunits A (coded by the *gyrA*) and B (*gyrB*), which encode a type II DNA topoisomerase, are associated with resistance to fluoroquinolones in TB [26]. High-level FQ resistance is conferred by mutations in subunit A and can lead to poor treatment outcomes in MDR/XDR-TB, a finding which was recapitulated in our study findings. Kanamycin (KM) inhibits protein synthesis through the modification of ribosomal structures at the 16S rRNA [27]. High-level resistance to injectable drugs (including kanamycin) can adversely impact the efficacy of the second-line treatment regimen for MTB. The *rrs* mutation at 1401G was shown to confer high-level kanamycin resistance and was also associated with higher patient mortality. The mutations in *ethA* are responsible for drug resistance to prothionamide (PTO), which is a group C drug of MDR-TB regimens. A study by Tan et al. reported that *ethA* (51.4%) was most frequently observed among PTO-resistant isolates, followed by mutations in the promoters of *inhA*, *inhA*, *ndh*, and *mshA*, which together confer increased resistance of MTB to PTO [28]. These second-line drugs have been used for the treatment of MDR-TB, and the emergence of drug resistance reduces treatment efficacy and increases the risk of poor treatment outcomes. The continued rise of drug resistance to TB is a looming threat to the prevention and control of the disease. WGS has significant potential to identify all known drug resistance mutations present in the genome and also possibly predict the level of drug resistance simultaneously. The WGS can overcome the limitations of conventional phenotypic DST and could provide an efficient personalized treatment regimen to individual patients and optimize their treatment outcomes [29].

The secondary findings from this systematic review were broad in scope and mostly involved laboratory aspects (e.g., correlations with phenotypic DST, cost, and turnaround time) or public health actions (e.g., determining transmission events within outbreaks). Both laboratory and public health aspects are important considerations for individual treatment (and subsequent outcomes) of TB cases. Compared with conventional phenotypic DST, WGS resistomes can be obtained within a relatively short turnaround time [17] and provide a substantially more comprehensive panel of polymorphisms associated with resistance compared with current commercially available tests for DR-TB. The detection and characterization of emerging drug-resistant mutations become the key to the appropriate treatment of TB and eradication of DR-TB infections. Novel mutations within known drug-resistance-associated genes were identified with WGS, and which were subsequently shown to be highly associated with phenotypic drug resistance [11,21,23], and mutations associated with low-level resistance were also characterized with unprecedented accuracy; however, none of the studies reported here included drug resistance associations with novel mutations leading to adverse outcomes.

WGS data have also aided in delineating information regarding transmission dynamics, as well as intra- and interpatient variations during outbreaks of TB infection. Epidemiological tracing of the spread of TB disease is an important factor in TB disease control. WGS has shown improvements in providing additional resolution in genotyping and transmission epidemiology compared with commercially available tools [30,31]. Makhado et al. identified the role of WGS in the transmission detection of the Ile491Phe-bearing lineage of the MDR tuberculosis strain in South Africa [22].

In recent years, Illumina has dominated the next-generation sequencing market with its sequencing-by-synthesis technique. Seven of our included studies used Illumina MiSeq or HiSeq platforms with varying degrees of throughput. It is important to note that different models or versions of sequencing platforms can result in significant variations in throughput, turnaround time, and associated sequencing costs. The HiSeq sequencer provides significantly higher data output and can process a large volume of samples, whereas the MiSeq instrument is a small-scale benchtop sequencer with a quicker turnaround time, making it suitable for small-scale clinical sample processing [32]. The MiSeq instrument has a lower initial cost compared with larger systems, such as HiSeq, but a higher cost per sequenced base due to its lower throughput. The other sequencing system used by one study in this review was the PyroMark Q96 ID system, which is a discontinued platform based on pyrosequencing and is more suitable for targeted gene sequencing. Overall, the establishment of WGS as a routine diagnostic tool can provide an in-depth understanding of the drug resistance, transmission, and genotype diversity of tuberculosis. This will accelerate the detection of drug-resistant TB and improve the patient’s treatment outcomes.

Some limitations of this study have to be acknowledged. First, the included studies were heterogeneous, precluding further meta-analysis. While most of the bioinformatics pipelines employed for WGS analysis were open source and freely available, the use of different tools and methods for data analysis was inconsistent across the studies included here, which can affect the calling and interpretation of drug-resistance-associated mutations. The studies included in this review did not report all TB drug-resistance-associated genes listed in the WHO guidelines [33]. As a result, there was a lack of correlation between mutations detected within drug-resistance-associated genes and treatment outcomes. There was also limited or no information on treatment regimens or drug combinations provided to the patients and any changes that were made to a patient’s treatment profile based on WGS resistome data were inconclusive. 

### Recommendations

We found that there is a scarcity of randomized control trials involving the use of WGS in TB treatment and the association of WGS resistome results with treatment outcomes. A broader investigative approach to prospectively include WGS is needed when designing future TB treatment studies to address the lack of evidence on the clinical effectiveness of WGS. 

A major issue in the field of TB-WGS is the wide range of bioinformatic analysis tools for the analysis and interpretation of drug resistance, disease transmission, and evolution. Multiple reasons are associated with the lack of standardized analyses, including access to appropriate facilities (e.g., computer systems, hardware, and data storage infrastructure) and a lack of trained personnel to perform testing and analysis protocols. The effect is a high degree of heterogeneity in results across different laboratories in different regions; such differences can lead to inconsistencies in interpretation and inferences of transmission. To overcome this, the establishment of a universally accepted TB-WGS gold standard for testing and analysis would ensure consistency and comparability across regions using different sequencing platforms and protocols and would provide a framework for WGS testing and analysis for other microorganisms. 

We found during this review that there was also inconsistent reporting on other factors associated with TB-WGS, including the cost of sequencing and turnaround time from sample receipt to result, and limited documentation on user understanding of WGS technology. While these details are not directly associated with treatment outcomes, such factors affect the utility of the TB-WGS results; for instance, prohibitive costs of sequencing and incomplete or lack of understanding of WGS technology and results may limit the uptake of routine TB-WGS, and as a result, hamper efforts to better understand the correlation between genomic and phenotypic results. Providing detailed descriptions of costs and ensuring the education of TB clinicians and healthcare professionals on the strengths and limitations of WGS can ensure that informed decisions can be made to enhance patient treatment and clinical outcomes. 

Robust, scalable pipelines and curated catalogs of drug resistance mutations are required for the fast and accurate analysis of WGS data and, ultimately, the recognition of drug resistance. In 2021, the WHO introduced a comprehensive catalog of drug-resistance-associated mutations, which serves as a benchmark for the interpretation of mutation-inducing resistance to first-line and second-line anti-tubercular drugs. The catalog can be used by TB laboratories throughout the world to aid in the interpretation of drug resistance mutations and assist in the development of new molecular drug susceptibility tests that help to correlate the association between genotypic mutations and phenotypic resistance [33].

## 5. Conclusions

Whole-genome sequencing presents an appealing alternative diagnostic approach in the era of emerging drug resistance in MTB. WGS has a significant capacity to provide comprehensive drug resistance data for MDR/XDR-TB, which enables personalized treatment selection to optimize treatment outcomes. Further studies are required to establish a uniform diagnostic algorithm for WGS that enables substantial changes in clinical decision making to provide a higher-quality health system.

## Figures and Tables

**Figure 1 pharmaceutics-15-02782-f001:**
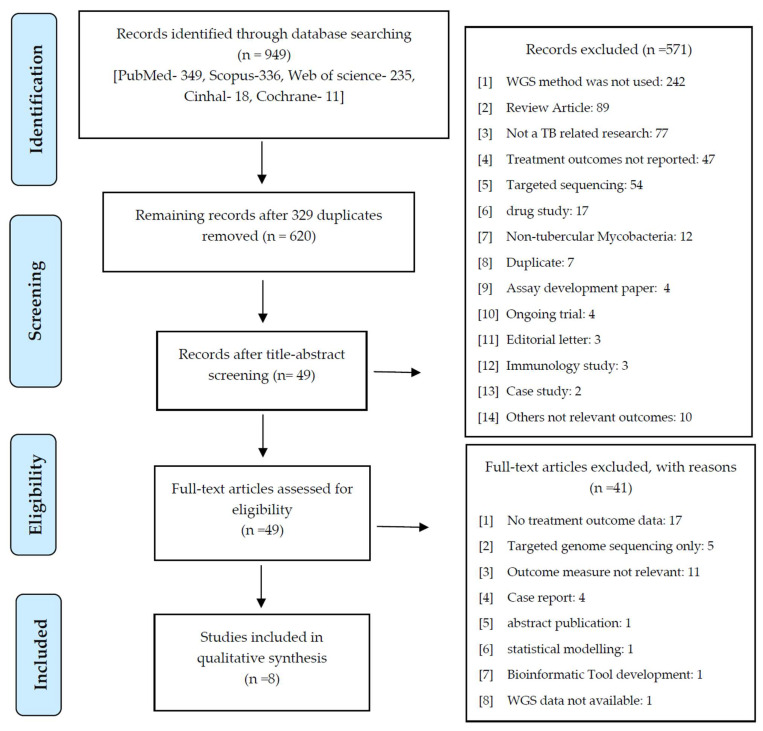
PRISMA flowchart of the selection process of the included studies.

**Table 1 pharmaceutics-15-02782-t001:** Characteristics of the included studies and relevant outcomes.

#	Author and Year	Country	Study Design	Study Period	Study Size	Sex (Male/Total)	Age ^$^ (*n*)	MDR-TB/XDR-TB Cases
1	Georghiou 2017 [19]	Moldova, India, South Africa	Observational cohort	April 2012–June 2013	451	296/451	<25 years: 90 25 to 49 years: 271 ≥50 years: 90	MDR/XDR-TB = 451
2	Makhado, 2018 [22]	South Africa	Retrospective observational	January 2013–September 2016	249	164/249	Mean age (years) ± SD: Wild-type: 39.2 ± 13.4 Ile491: 35.4 ± 11.3	MDR-TB = 37
3	He 2020 [17]	China	Prospective observational	January 2014–September 2016	123	108/123	Mean age: 43.5 ± 13.6	MDR-TB = 123
4	Clark 2013 [20]	Uganda	Cohort study	July 2003–April 2007	41	17/29	Mean age: 32.3 years ± 8.8	MDR-TB = 41
5	Lempens 2020 [23]	Bangladesh	Retrospective cohort	March 2005–March 2015	449	316/449	10 to <20 years: 47 20 to <30 years: 152 30 to <40 years:104 40 to <50 years: 75 50 to <60 years: 45 ≥60 years: 26	MDR/RR-TB = 449
6	Katale 2020 [11]	Tanzania	Cohort study	2014	87	40/87	Median age: 35 years (range: 29–44)	MDR-TB = 24, mono-resistant-TB = 17, polyresistant TB = 16
7	Chen 2020 [24]	China	Retrospective observational	January 2014–September 2016	94	72/94	<60 years: 88 >60 years: 6	MDR-TB = 42/94, pre-XDR R-SLID = 33/94, pre-XDR R-FQ = 16/94, XDR-TB = 3/94
8	Zürcher 2021 [21]	Côte d’Ivoire, Democratic Republic of the Congo, Kenya, Nigeria, Peru, South Africa, Thailand	Cohort study	September 2014–July 2016	582	357/582 (MDR-TB = 90/146 XDR-TB = 11/24)	Median age: 33 years (range: 27–43) MDR-TB = 31 years (range: 25–39) XDR-TB = 30 years (range: 25–34)	MDR-TB = 146, XDR-TB = 24, mono-resistant-TB = 35, other poly-resistant-TB = 38

#—represents study number ^$^—Age measurements for studies 1, 5, and 7 were provided as age ranges only, studies 2, 3, and 4 were provided as mean age + standard deviation, and studies 6 and 8 were provided as median age (range).

**Table 2 pharmaceutics-15-02782-t002:** Additional population characteristics of studies included in this review.

#	HIV (n/Total)	Diabetic (n/Total)	TB Treatment Status (n/N)	Sequencing Technology	pDST Method	Follow-Up Period (Months)	Successful Treatment Outcomes [Cure/Treatment Completion]	Unfavorable Treatment Outcomes [Death/Treatment Failure/Other]
1	68/451	21/451	New 135/451, prior treatment 316/451	PyroMark Q96 ID system	MGIT 960	11.96	Total with outcomes: 363/451 (80.5%)	Total with outcomes: 88/451 (19.5%) Death: 88
2	28/37	Not specified	Prior treatment 25/37	Illumina Hiseq 2000	MGIT 960	6	Total with outcomes: 9/37 (24.3%) Cure: 3 Treatment completion: 6	Total with outcomes: 19/37 (51.3%) Death: 3 Treatment failure: 10 Defaulted treatment: 6
3	Not specified	97/123	Prior treatment (2nd line) drugs 43/123	Illumina Miseq or X10	LJ media and MGIT 960	Not specified	Total with outcomes: 74/123 (60.1%) Cure: 67 Treatment completion: 7	Total with outcomes: 49/123 (39.8%) Death: 5 Treatment failure: 30 Loss to follow up: 14
4	11/41, unknown HIV status 3/41	Not specified	Recurrent TB or relapse MDR-TB: 41	Illumina Hiseq	MGIT 960	Patients followed up until death or end of 2006	Total with outcomes: 1/41 (2.4%) Cure: NA Treatment completion: 1	Total with outcomes: 4/41 (9.7%) Death: 4
5	Not specified	Not specified	Not specified	Illumina HiSeq or MiSeq	LJ media and Middlebrook 7H11 agar	Not specified	Total with outcomes: 356/449 (79.3%) Cure: 344 Treatment completion: 12	Total with outcomes: 93/449 (20.7%) Death: 27 Treatment failure: 19 Relapse: 8 Loss to follow up: 39
6	20/87	19/87	New 45/87, prior treatment 42/87	Illumina MiSeq	LJ media	Not specified	Total with outcomes: 47/87 (54.0%) MDR-TB 17, mono-resistance-TB 16, polyresistance TB 14	Total with outcomes: 10/57 (17.5%) Death: 3 (MDR 2, polyresistant TB 1) Treatment failure: NA Defaulted treatment: 7 (MDR 5, mono-resistant TB 1, polyresistant TB 1)
7	Not specified	31/94	Prior treatment 68/94	Illumina Miseq or X10	LJ media and MGIT 960	27	Total with outcomes: 74/94 (78.8%)	Total with outcomes: 20/94 (21.2%) Death: NA Treatment failure: 20
8	247/582 (MDR 43/146, XDR 10/24)	Not specified	History of TB 209/582 (MDR 90/146, XDR 19/24)	Illumina HiSeq 2500	LJ media and MGIT960	Not specified	Total with outcomes: 377/582 (64.8%) (MDR 76, XDR 11)	Total with outcomes: 205/582 (35.2%) Death: 64 (MDR 24, XDR 9) Treatment failure: 21 (MDR 5, XDR 2) Transfer out of the program: 28 (MDR 10, XDR 2) Loss to follow up: 55 (MDR 22) Study ongoing/outcome indeterminate: 37 (MDR 9)

Note: #—represents study number—see Table 1 for Author and publication year details; MDR-TB—multidrug-resistant tuberculosis; RR-TB—Rifampicin-resistant tuberculosis; XDR—extensively drug-resistant tuberculosis; SLID—second-line injectable drugs; FQ—Fluoroquinolones; LJ media—Lowestein–Jensen media; MGIT960—Mycobacterial growth indicator tube 960.

**Table 3 pharmaceutics-15-02782-t003:** Additional relevant findings for studies included in this review.

#	Additional Relevant Findings
1	High-level fluoroquinolone (FQs) resistance (*gyrA* 94AAC and 94GGC mutations) (OR, 3.99 [95% CI, 1.10 to 14.40]) and kanamycin (*rrs* 1401G mutation) (OR, 5.47 [95% CI, 1.64 to 18.24]) were significantly associated with patient mortality.
2	Mutation in *rpoB* Ile491Phe of MDR-TB was related to patient mortality. WGS enables the detection of the transmission of MDR-TB with *rpoB* Ile491Phe mutation, which remains undetected when using routine diagnostic tools. WGS measured the genomic distances based on SNP variation and showed that two lineages (4.4.1.1 and 4.1.1.3) of *rpoB* Ile491Phe had emerged independently.
3	Patient mortality was correlated with WGS detection of the following: - *gyrA* mutation conferring high-level fluoroquinolone resistance (OR, 3.947; 95% CI, 1.195–13.034; *p* = 0.024); - *ethA* mutation conferring prothionamide resistance (OR, 3.817; 95% CI, 1.154–12.823; *p* = 0.028); - Mutations high-level moxifloxacin phenotypic resistance (OR, 4.362; 95% CI, 1.364–13.950; *p* = 0.013); - Mutations conferring cycloserine phenotypic resistance (OR, 7.457; 95% CI, 1.644–33.819; *p* = 0.009). Overall WGS costs ($USD63), for each patient and the average turnaround time (7 days) were advantageous to culture-based DST.
4	Longitudinal sampling of TB cases with recurrent or relapse TB demonstrated the acquisition of drug resistance-associated SNPs. WGS identified 8 patients spanning three clusters, with almost identical genetic profiles which suggests transmission of multidrug-resistant disease.
5	High-level fluoroquinolone resistance was associated with a clinically adverse outcome compared with fluoroquinolone-susceptible TB (aOR 10.3; 95% CI 3.1–35.0; *p* < 0.001). High-level isoniazid resistance predicted treatment failure on either phenotypic DST (aOR 3.8; 95% CI 1.03–13.7; *p* = 0.04) or a combination of phenotypic DST and genotypic DST (aOR 3.8; 95% CI 1.03–13.7; *p* = 0.04).
6	Strong agreement was reported between WGS and phenotypic DST for rifampicin (97%), isoniazid (81%), and streptomycin (95%), as well as with Xpert MTB/RIF for the detection of rifampicin resistance (95%) (kappa = 1.00).
7	Fourteen (77.8%) TB cases developed acquired drug resistance under ineffective treatment. The insufficient number of effective drugs in the combined treatment regimen was the main reason for MDR-TB treatment failure. WGS detected low-frequency resistance mutations and heterogeneous resistance with high sensitivity.
8	The concordance between conventional phenotypic DST and WGS was 80% for pan-susceptible, 8% for mono-resistant, 66% for MDR, and 33% for pre-XDR or XDR tuberculosis. Based on the WGS resistome results, 15% (77/530) of study participants were treated inappropriately, among which 11% (60/530) were undertreated and 3% (17/530) were overtreated. The odds of death in undertreatment TB cases were 4.92 (95% CI 2.47–9.78) compared with patients receiving appropriate treatment.

#: represents study number—refer to Table 1 for author and publication year details; OR—odds ratio; aOR—adjusted odds ratio; 95% CI—95% confidence interval.

**Table 4 pharmaceutics-15-02782-t004:** NHLBI Quality Assessment Tool for Observational Cohort and Cross-Sectional Studies.

Criteria	Georghiou [19]	Makhado [22]	He [17]	Clark [20]	Lempens [23]	Katale [11]	Chen [24]	Zürcher [21]
1. Was the research question or objective in this study clearly stated?	Yes	Yes	Yes	Yes	Yes	Yes	Yes	Yes
2. Was the study population clearly specified and defined?	Yes	No	Yes	Yes	Yes	Yes	Yes	Yes
3. Was the participation rate of eligible persons at least 50%?	Yes	CD	Yes	Yes	Yes	Yes	Yes	Yes
4. Were all the subjects selected or recruited from the same or similar populations (including the same period)? Were inclusion and exclusion criteria for being in the study prespecified and applied uniformly to all participants?	Yes	NR	Yes	No	Yes	Yes	Yes	Yes
5. Was a sample size justification, power description, or variance and effect estimates provided?	NR	NR	NR	NR	NR	NR	NR	NR
6. For the analyses in this study, were the exposure(s) of interest measured prior to the outcome(s) being measured?	Yes	Yes	Yes	Yes	Yes	Yes	No	Yes
7. Was the timeframe sufficient so that one could reasonably expect to see an association between exposure and outcome if it existed?	Yes	NR	Yes	NR	NR	NR	Yes	NR
8. For exposures that can vary in amount or level, did the study examine different levels of the exposure as related to the outcome (e.g., categories of exposure, or exposure measured as a continuous variable)?	Yes	Yes	Yes	NR	Yes	NR	Yes	Yes
9. Were the exposure measures (independent variables) clearly defined, valid, reliable, and implemented consistently across all study participants?	No	Yes	Yes	Yes	No	Yes	Yes	Yes
10. Was the exposure(s) assessed more than once over time?	No	Yes	No	Yes	No	No	Yes	No
11. Were the outcome measures (dependent variables) clearly defined, valid, reliable, and implemented consistently across all study participants?	Yes	Yes	Yes	No	Yes	Yes	Yes	Yes
12. Were the outcome assessors blinded to the exposure status of participants?	NR	Yes	NR	NR	NR	NR	NR	NR
13. Was the loss to follow-up after baseline 20% or less?	Yes	Yes	Yes	CD	Yes	Yes	Yes	Yes
14. Were key potential confounding variables measured and adjusted statistically for their impact on the relationship between exposure(s) and outcome(s)?	Yes	Yes	Yes	No	Yes	CD	No	Yes
**Quality rating (good, fair or poor)**	** Fair **	** Fair **	** Good **	** Poor **	** Fair **	** Poor **	** Fair **	** Fair **

Note: CD, cannot determine; NR, not reported.

## Data Availability

Most included studies are publicly available via open access on journal websites. Additional data and code are available upon request.

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
