# Peer review of "Impact of Whole-Genome Sequencing of Mycobacterium tuberculosis on Treatment Outcomes for MDR-TB/XDR-TB: A Systematic Review"

_pharmaceutics, 2023, doi:10.3390/pharmaceutics15122782_

Round 1

Reviewer 1 Report

Comments and Suggestions for Authors

Comments to the Author:

The manuscript titled “Impact of whole-genome sequencing of Mycobacterium tuberculosis on treatment outcomes for MDR-TB/XDR-TB: A Systematic Review” offers a comprehensive and insightful exploration of the role that whole-genome sequencing (WGS) of Mycobacterium tuberculosis plays in managing and treating multi-drug resistant tuberculosis (MDR-TB) and extensively drug-resistant tuberculosis (XDR-TB). The authors should be commended for their meticulous approach to the experimental design and analysis, which culminates in a well-rounded assessment of the subject matter. Despite these strengths, there are a few aspects of the manuscript that could benefit from further elucidation and enhancement. Detailed comments to elaborate on these points are provided below.

Major:

1.        In lines 151-159, the authors are kindly advised to endeavor to provide an in-depth description of the specific models of Illumina sequencing platforms employed in the various studies under discussion. The discrepancies in models or versions of these sequencing platforms can lead to significant variations in throughput, turnaround time, and associated sequencing expenses. Providing such detailed information, if it is readily available, would immensely benefit the readers by granting them a more explicit and holistic comprehension of the experimental conditions. This, in turn, could significantly enhance the interpretability of the study’s results, thereby adding value to the overall quality of the manuscript.

2.        In the section “4.1. Recommendations”, I suggest adding a discussion on a pressing issue currently faced by the WGS data analysis field: the lack of a unified and internationally recognized standard for WGS data analysis. This absence has led to a high degree of heterogeneity in the results derived from different analysis pipelines, making effective comparison and assessment challenging. Addressing this issue is of utmost urgency as it directly impacts the reliability and accuracy of WGS sequencing outcomes. We urgently need to establish industry consensus and promote the development of a universally accepted gold standard for WGS data analysis. This will enhance the consistency and comparability of results, laying a solid foundation for the future applications of WGS sequencing across various domains.

3.        In assessing the influence of Mycobacterium tuberculosis Whole Genome Sequencing (WGS) on treatment outcomes for Multi-Drug Resistant Tuberculosis (MDR-TB) and Extensively Drug-Resistant Tuberculosis (XDR-TB), it is crucial to holistically consider various factors. This includes not just the relationship between WGS and phenotypic Drug Susceptibility Testing (DST), but also aspects such as cost, turnaround time, and the technology’s user-friendliness. This latter point is particularly paramount for practitioners such as hospital physicians, who may lack specialized training in genomic analysis. Ensuring that WGS tools and platforms are accessible and intuitive for these healthcare professionals is essential. Only by doing so can we guarantee the effective integration and exploitation of this technology within clinical practices, optimizing patient outcomes and advancing tuberculosis treatment.

Minor Comments:

1.        In Line 84 of the manuscript, I have noticed subtle discrepancies in the use of certain terminology. Specifically, an abbreviation is established for a specialized term, yet its full form is subsequently utilized, as evidenced by “multidrug-resistant/extensively drug-resistant tuberculosis”. To enhance the manuscript's clarity and coherence for readers, I strongly recommend maintaining consistency in the application of abbreviations throughout the text.

2.        Line 100: Please ensure uniformity in your use of quotation marks for key terms. The termsdrug-resistant tuberculosis”, ‘tuberculosis’, treatment outcome”, and whole genome sequencing” are inconsistently presented with both double and single quotation marks. For clarity, it would be advisable to maintain a consistent style throughout.

3.        Line 158: The gene namerpoB” should be formatted in italics. Furthermore, I have noticed multiple occurrences within the manuscript where gene names, which are conventionally presented in italicized text in scientific literature, have not been consistently formatted as such. I strongly recommend that the authors conduct a comprehensive review of the manuscript to identify and correct any inconsistencies in the formatting of gene names, ensuring uniformity and alignment with standard scientific conventions.

4.        In Section “3.2. Whole Genome Sequencing and Analysis of Drug Resistance Mutations” (Line 150), the manuscript utilizes the complete phrase "Whole Genome Sequencing." Contrastingly, in Section “3.4. WGS and Detection of Undertreated TB Cases” (Line 211), the abbreviation “WGS” is employed. To enhance consistency and clarity throughout the document, I recommend standardizing the use of terminology across all sections.

5.        Line 224: There is an extraneous period at the start of the paragraph. Kindly review and correct this typographical oversight to enhance the clarity and coherence of the manuscript.

6.        It would be beneficial to enhance the quality of the figures for the final version to ensure optimal clarity and visual appeal.

7.        Table 2 is missing crucial footnotes, particularly for the terms “CD” and “NR”, which require further clarification.

Author Response

Dear reviewer,

We thank you for your time and suggestions to improve this manuscript. Please see attached word document for our detailed responses.

Kind regards,

Connie Lam, Druti Hazra

Reviewer 2 Report

Comments and Suggestions for Authors

*Title: Impact of Whole-Genome Sequencing of Mycobacterium tuberculosis on Treatment Outcomes for MDR-TB/XDR-TB: A Systematic Review*

*Introduction:*

Lines 44-45: The author cites epidemiological data on TB using 2019 data (4 years old). Update the data.

Lines 60-63: Initially, the author presents data on the PCR technique and then, after discussing the limitations of the technique, discusses the advantages of sequencing. The use of an adversative conjunction to separate the paragraphs is missing (“however, nevertheless, on the other hand”).

Lines 74 and 76: The author presents different studies on the topic, but it is necessary to better explain their contributions and not just comment on the type of study they conducted.

At the end of the introduction, it lacks a more explicit justification, explaining why this review was conducted. "Due to the fact that..., new studies need to be developed..., in this sense, this review seeks... Make clear the academic/scientific contribution of this review."

*Materials and Methods:*

Lines 127 and 128: The author mentions 14 quality parameters used, but does not list these parameters. List them briefly. Alternatively, it can be indicated that the parameters are listed in Table 2 in the results. But this information needs to be noticeable to the reader from the beginning.

*Results:*

Line 152 with contrasting information with line 137: in line 137, the author states that a total of 8 studies were used, while in line 152, the total is 7. Clarify how the 8th study was excluded from the research. Between lines 160 and 161, there is again a total of 8 studies analyzed.

Improve the quality (resolution) of Figure 1.

*Discussion:*

Overall, the data presented in the discussion need to be compared with each other and not just presented (as done in the results). Make better use of conjunctions to improve the coherence of the text. For example, "On the other hand...", "In agreement...".

Author Response

Dear reviewer,

Thank you for your time and helpful suggestions to improve this manuscript. Please see attached word document for our detailed responses.

Kind regards,

Connie Lam, Druti Hazra

Reviewer 3 Report

Comments and Suggestions for Authors

In my opinion, the article fulfils the criteria imposed by PRISMA to be published. Although the topic is well known and the role of WGS is well established, I think the information provided by the authors is important for clinicians. I have a question for the authors: How do you explain that out of over 900 studies, only 8 were selected? Were the selection criteria too strict? Also, how do you explain the lack of studies from the USA or Eastern Europe?

Author Response

We thank the reviewer for their time, valuable comment and suggestion to improve this manuscript and would like to respond the reviewer's comments as follows:

The main goal of this systematic review was to determine whether the use of WGS in diagnosis or treatment of tuberculosis cases was associated with treatment outcomes. As a result, the two main criteria for inclusion in this study was a) the use of WGS technology AND b) the inclusion of treatment outcomes. During the course of our review, we found that while there were many studies utilizing WGS technology in tuberculosis cases however very few studies also included treatment outcomes. We restricted our review to include MDR/XDR-TB patients as these cases were likely to benefit most from the detection of drug resistance associated mutations using WGS, however there is potential to extend this in future to include susceptible TB cases.

Despite the high uptake of WGS technology in the USA and Western Europe, the incidence of MDR/XDR TB in these regions is relatively low; in contrast, the incidence of TB in Eastern Europe is comparatively higher, but uptake of WGS technology is substantially lower in this region. As a result, very few studies included the desired patient cohort, and we were unable to find any that included treatment data which were not specific case studies (which was an exclusion criterion).

We do note that the 8 studies identified in this review are mainly from countries where the TB incidence is high, and/or were multi-center studies which included one or more high incidence TB countries. Future multi-center studies are required to better represent worldwide trends on the use of WGS as a diagnostic tool and its impact on treatment outcomes.

Round 2

Reviewer 1 Report

Comments and Suggestions for Authors

All questions have been addressed.

Reviewer 2 Report

Comments and Suggestions for Authors

The authors have modified the manuscript as suggested.